# Surface and Subsurface Damage Caused by Bullet Impacts into Sandstone

**Oliver Campbell** [1],*[ID], **Tom Blenkinsop** [1][ID], **Oscar Gilbert** [2] **and Lisa Mol** [2]

1    School of Earth and Environmental Sciences, Cardiff University, Cardiff CF10 3AT, UK;
     BlenkinsopT@cardiff.ac.uk
2    Department of Geography and Environmental Management, University of the West of England,
     Bristol BS16 1QY, UK; Oscar.Gilbert@uwe.ac.uk (O.G.); Lisa.Mol@uwe.ac.uk (L.M.)
*    Correspondence: campbellor@cardiff.ac.uk

**Abstract:** The shift of armed conflicts to more urbanised environments has increased the risk to cultural heritage sites. Small arms impacts are ubiquitous in these circumstances, yet the effects and mechanisms of damage caused are not well known. A sandstone target was shot under controlled conditions to investigate surface and subsurface damage. A 3D model of the damaged block, created by structure from motion photogrammetry, shows that internal fracturing was at least as extensive as the visible surface fractures. Backscatter electron imaging of the damaged surface shows a shift from intragranular fracturing and grain size reduction at <5 mm from the impact point to primarily circumgranular fracturing and grain 'plucking' at 20 mm from the impact point. Internal fracture intensity decreased with distance from the centre of the crater. Volumes around the impact point are therefore at greater risk of subsequent weathering deterioration, but significant damage extends to the periphery of the target, rendering whole blocks vulnerable. The surface crater, despite being one of the most conspicuous aspects of conflict damage, has many times less area than internal and surface fractures.

**Keywords:** bullet damage; fracture analysis; microstructures; photogrammetry; heritage; sandstones

## 1. Introduction

Loss of life, civilian displacement, and damage to property are inevitable consequences of armed conflicts. As modern conflicts shift towards more urbanised environments, the chance of damage to cultural property, defined here as tangible heritage (e.g., sites, buildings, and artefacts), increases [1]. Ideological extremism is a driver of intentional demolition of cultural property, a tactic infamously employed by Islamic State (IS)/Da'esh. Historic sites such as Palmyra, Mosul, and Nimrud made media headlines after IS propaganda videos were released showing the use of sledgehammers, bulldozers, and explosives to cause damage [2]. Further, collateral damage may be caused by airstrikes and artillery, such as the severe damage to Sana'a in Yemen by Saudi airstrikes [3]. This wide spectrum of damage sources has culminated in the harm to, or loss of, many heritage sites across the Middle East and North Africa region.

Within this spectrum of damage, albeit on a smaller scale, is damage caused by the widespread use of small arms within current conflicts. Impact damage from bullets and shrapnel is under-researched, although initial studies show small arms' impacts increase the long-term deterioration of stone [4–6]. Impacts cause compaction and grain size reduction near the point of contact, causing relatively less surface hardness reduction than surrounding regions. Surrounding regions also exhibit increased surface permeability, suggesting greater susceptibility to the ingress of weathering agents such as moisture and salt [5–7]. Moisture can act to dissolve matrix minerals and cement in the stone, loosening grains, increasing porosity, and reducing overall strength [8–10]. Meanwhile, precipitation

of salts from solution forces grains apart, further weakening the stone [11]. The development of fracture networks increases the depth within the stone to which these processes can extend, expanding the region at risk of deterioration [12]. Measurement of such effects is vital in assessing portions of heritage at the highest risk of further deterioration.

In situ measurement of stone properties is therefore highly desirable for heritage conservation efforts but is generally restricted to non-destructive testing [13–16]. Field instruments such as surface hardness probes, permeameters, ultra-pulse velocity meters, moisture probes, and infra-red scanners can provide valuable information on stone conditions, but they cannot be used safely in current conflict areas [5,7,17]. The non-destructive nature of these methods precludes direct observation of subsurface damage, for which alternative methods are required.

Controlled experiments, such as those simulating meteorite impacts, are one possibility. Meteorite impact simulations can target natural stone, and use destructive methods such as thin sectioning to study subsurface damage [18–20]. However, these studies typically use spherical, single composition projectiles and have impact velocities exceeding 1.5 kms$^{-1}$, whereas small arms projectiles are typically ogive-nosed, composed of multiple materials, and have velocities in the range of 0.5–1.0 kms$^{-1}$. Beyond engineering focussed studies of ceramic and metal plate targets, few experiments exist which can provide insights into the effects of projectiles fired by small arms [21,22]. Gilbert et al. [7] studied the effects of bullet impact on the surface hardness and surface permeability of sandstone. Non-destructive testing on the stone surface highlights areas of increased permeability and decreased hardness, with the greatest permeability increases associated with large radial fractures.

This study extends the work of Gilbert et al. [7] by characterising the surface morphology of impact damage and quantifying macro-scale fracture networks using 3D models generated by photogrammetry. It describes the microscale surface damage within the crater using electron microscopy and highlights the link to subsurface damage observed through thin section microscopy and fracture intensity analysis.

## 2. Materials and Methods

### 2.1. Target Stone and Projectile Properties

A cube of sandstone (14.7 × 14.7 × 14.7 cm) was quarried from the Huesca region of Northern Spain because of its analogous properties to heritage stones in the Middle East, such as the Umm Ishrin sandstones at Petra in Jordan, which have sustained historical damage [23,24]. The Umm Ishrin is a well-sorted, medium-coarse-grained sandstone comprised of quartz grains and varying amounts of calcite and clay cement [24]. The Huesca sandstone is a well-consolidated, medium-grained sandstone with an average pore size of 40–70 μm. X-ray diffraction (XRD) analysis revealed a composition of quartz and calcite, with lithic fragments and matrix comprised of clay minerals (muscovite, kaolinite and clinochlore) (Figure 1b,e) [25]. Thin section observations of undamaged Huesca sandstone show no inherent fractures and no apparent anisotropy at the scale of the sample, showing that the fractures described here were caused by the bullet impact (Figure 1e) and not inherited. The block was shot with 7.62 × 39 mm ammunition, typical of many Kalashnikov (AK) variant rifles, including the well-known AK-47, used widely in past and contemporary conflicts. It was fired from an AK-103 rifle at a range of 200 m, resulting in a velocity (*v*) of ~540 ms$^{-1}$ at the point of impact. The projectile is constructed from a brass jacket and lead core, with a spitzer ogive-nose shape and has a mass (*m*) of 7.95 g (123 grains), resulting in a kinetic energy (K$_E$ = 0.5 $mv^2$) upon impact of 1.168 kJ (Figure 1a).

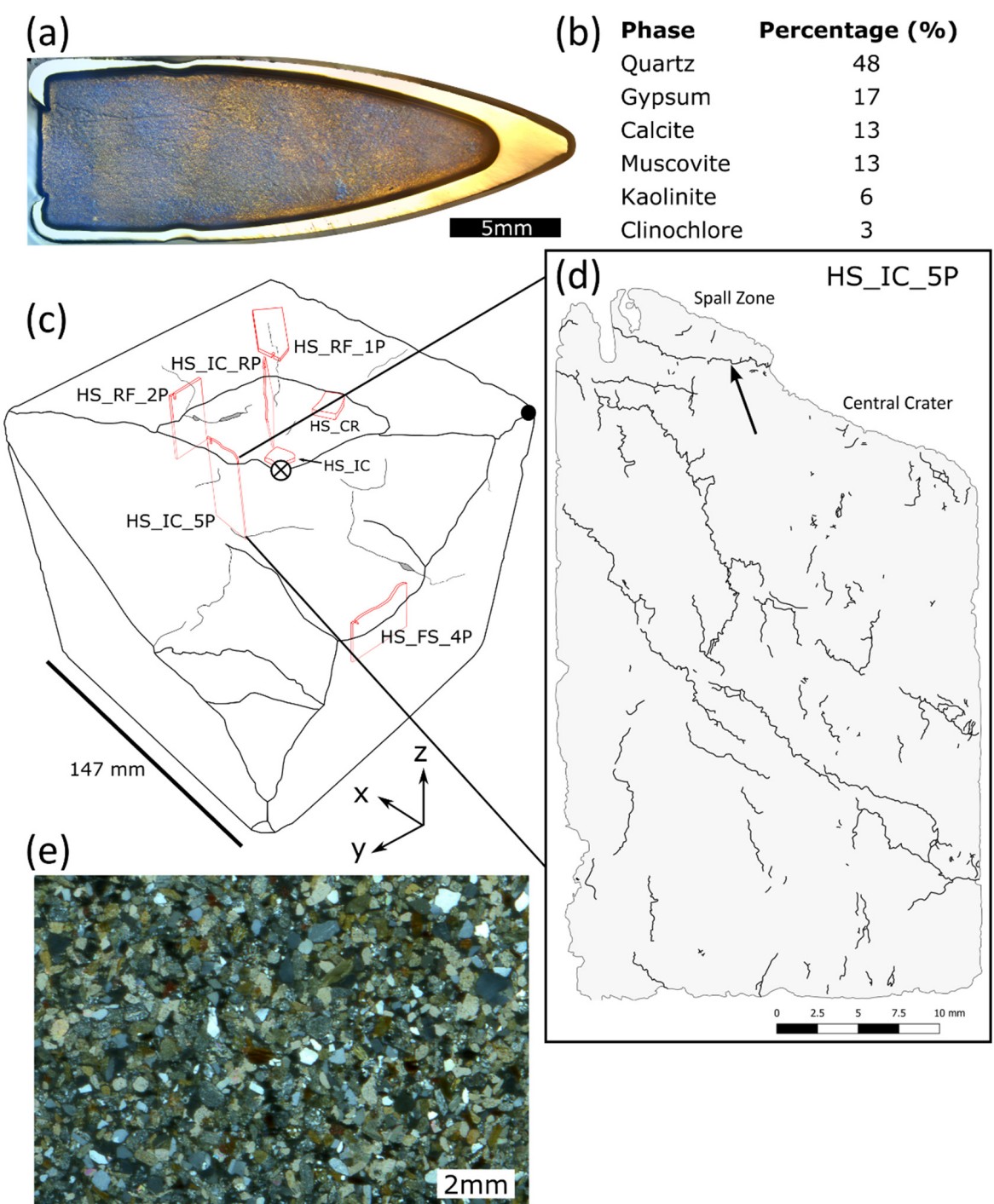

**Figure 1.** (**a**) Reflected light micrograph of a cross-section through a typical 'soft core' 7.62 × 39 mm projectile. The outer brass jacket surrounds the grey lead core. (**b**) Summary table of constituent minerals in the Huesca sandstone (taken from [25]). (**c**) Schematic figure of the Huesca sandstone block after being shot with 7.62 × 39 mm ammunition from a range of 200 m. Red outlines indicate the position and orientation of thin sections taken from within the sample. Crossed circle marks the centre of the crater. Solid circle indicates origin of 3D coordinate scheme. (**d**) Digitised fracture network from sample HS_IC_5P used in NetworkGT to calculate $P_{xy}$ values. Black arrow indicates a spall fracture below an incipient spall fragment (Complete fracture maps and transmitted light micrographs of each sample are available in Supplementary Information Figures S1–S5). (**e**) Transmitted light thin section micrograph of undamaged Huesca sandstone taken under cross polarised light.

### 2.2. Characterising Damage Morphology

A 14-megapixel Fujifilm FinePix S3400 (Tokyo, Japan) digital camera was used to photograph the sample through a 360° rotation at three overlapping camera positions. The sample was then overturned and the process was repeated. Additional images were taken of the damaged surface to ensure adequate capture of morphology. One hundred and forty-two images were imported into Meshroom (v2020.1.1), a free and open-source structure from motion (SfM) pipeline developed by AliceVision® [26,27]. The resultant 3D-textured mesh was scaled and oriented in 3D space using CloudCompare (v2.11.3) [28].

The FACETS plugin for CloudCompare [29] was used to summarise the morphology of impact damage. A Kd-tree algorithm was selected to summarise the model because of its faster processing time and better representation of geometry than the alternative fast-marching algorithm. The following settings were used in the Kd-tree: max angular difference = 5°, max relative distance = 1.00, max distance at 99% = 0.2, min points per facet = 10, and max edge length = 0.30. Facets representing undamaged areas of the block were manually removed. These settings were chosen to represent the damage adequately within workable processing timeframes (minutes vs hours). A smaller angular difference would have represented the morphology with a greater number of facets and complexity, but the increased processing time and larger data set had a negligible influence on the clustering observed in the stereonet.

The Compass plugin was used to digitise surface fracture traces and estimate their orientations [30]. The 3D mesh and digitised fracture traces were then imported into Blender (v2.92.0) [31] to estimate the minimum internal surface area of fractures. In order to compare these values with the areas of damage at the surface, the scaled and oriented model was imported into Meshlab where the surface fracture area was calculated [32,33]. The volume of material removed from the damaged block was also calculated in Meshlab.

Fracture planes from manual tracing (n = 24) and facet extraction (n = 674) are presented on standard equal area lower hemisphere projections (Figure 2). Facet data was contoured using a modified Kamb method with exponential smoothing [34,35]. The Kamb contour method was chosen over alternatives, such as the 1% area, because it is independent of sample size.

### 2.3. Microscale Analysis

Two stubs (~10 × 10 mm) were cut from the impact crater and coated with a 30 nm thick Au-Pd coating for use in a scanning electron microscope (SEM). Backscatter electron (BSE) images were obtained using an FEI (Cambridge, UK) Quanta FEG 650 with an Oxford Instruments (Abingdon, UK) Xmax$^n$ EDS detector. Images were captured at pressure with a spot size of 5.0, a working voltage of 5.00 kv, and a working distance of 8.5–11.6 mm. Thin sections (28 × 48 mm) were cut from different regions of the damaged block, with section planes oriented perpendicular to visible fractures (Figure 1c). To locate sections and damage within the block, a 3D coordinate scheme adapted from Tikoff et al. [36] was used. The target face of the sample is the XY plane and the Z axis is parallel to the bullet trajectory and negative into the block. The crater centre is used as the reference point for all distance measurements and is the point on the current crater floor that is directly below the point of impact.

Thin sections were scanned using an Epson (Amsterdam, Netherlands) Perfection 3170 photo scanner at 6400 dpi under plane and cross polarised light. Fractures were digitised in QGIS (v3.16.0) as a single polyline to preserve fracture geometry and characteristics (an example is shown in Figure 1d). Important characteristics of fracture networks, such as length and orientation, can differ between interpretations conducted by different investigators [37,38]. Analysing fracture branches instead of full traces reduces this bias, as well as mitigating any censoring effects of the sample region because the intersection with the edge now only affects a single branch, instead of the full fracture trace [37]. The NetworkGT plugin for QGIS was used to calculate $P_{xy}$ values for each thin section [39]. $P_{xy}$ values characterise fracture frequencies, intensities, and volumes, where *x* represents

the dimension of the sampling region and $y$ is the dimension of measurement [40,41]. For example, $P_{21}$ is a measure of fracture length ($L$) per area ($A$):

$$P_{21} = \sum L/A \tag{1}$$

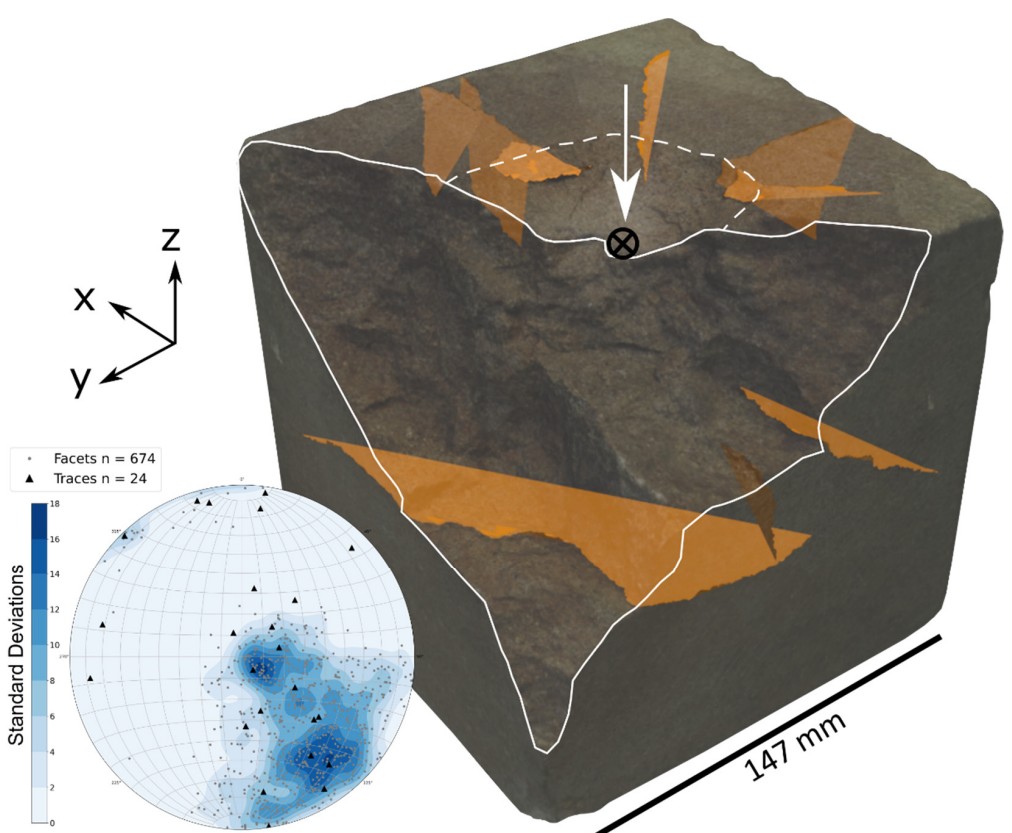

**Figure 2.** Summary of data measured from the 3D model of Huesca sandstone shot with 7.62 × 39 mm ammunition. A rendering of the block is visible with the minimum extent of internal fracturing estimated from surface traces shown in orange. Note the large fracture just below the centre of the model that is sub-parallel to the target face (overview 3D model is available in Supplementary Information Figure S6). The stepped damage region is outlined by solid white, and the impact crater by a dashed white line. The white arrow shows the bullet trajectory and black crossed circle marks the crater centre. (inset) A lower hemisphere equal area projection of the poles to fractures estimated from surface traces (black triangles), and the orientation of facets (grey circles) representing the stepped morphology of the damage surface. The facet orientation data is contoured in blue using a modified Kamb contour, indicating two distinct clusters of orientations: A steep NW dipping set and sub-horizontal set.

This per length ($L^{-1}$) unit is defined as fracture intensity and can be scaled to 1- and 3-dimensions. Dimensionless intensity values are those where the dimension of measurement and sampling are the same (e.g., $P_{22}$) [37]. $P_{22}$ values are calculated by the equation:

$$P_{22} = P_{21} \cdot L_c \tag{2}$$

where $L_c$ is the characteristic length, defined simply as the arithmetic mean of branch lengths [37]. The minimum $P_{32}$ value of the damaged block was calculated using the 3D model and Blender derived internal fractures:

$$P_{32} = A_f/V \tag{3}$$

where $A_f$ is the sum of surface and internal fracture area and $V$ is the volume of the damaged block derived from the 3D model.

The centre of the crater represents the point directly below the impact, so is used as the reference location from which sample distances are measured. Uncertainty in the distance from the crater centre measurements is estimated to be $\pm$ 2 mm, based on the contribution of several factors: (i) The measurement of section locations during the cutting process. (ii) The possible loss of material at the edges of thin sections during production, though every effort was made to minimise this. (iii) The scaling of the 3D model. (iv) The measurement of points on the 3D model. The digitisation in QGIS was the primary source of uncertainties in the calculation of fracture intensities. The optical thin-section scans used for digitisation are limited in their resolution at high magnifications. Despite a very high resolution of scanning (6400 dpi), grain boundaries and fracture edges are not sharp. The averaging of colour values across pixels in an image mean boundaries appear gradational at high zoom levels. For the lateral placement of polylines, important in determining the sample area, this uncertainty was individually estimated for each section, with values between 0.029 and 0.033 mm. The perimeter of the measured sample area was then expanded and contracted by these uncertainties to determine the maximum and minimum sample areas respectively.

There is a level of uncertainty in digitising the end-point of fractures along grain boundaries. At the scale of observation, fracture apertures can narrow to the point they become indistinguishable from the gradient of adjacent grain boundaries. In this situation, fracture trace was terminated if there was no distinguishable aperture when it reached grain boundaries, or there was no clear continuation of the fracture beyond that grain. An uncertainty of 0.1 mm was deemed appropriate as it is approximately 3–4 times the measured 'gradients' in boundary locations, so represents an average combined uncertainty where multiple grains are in contact. A minimum and maximum fracture trace network was calculated by decreasing and increasing the length at 'I' nodes by this uncertainty. The maximum $P_{21}$ and $P_{22}$ values were calculated using the minimum sample area and maximum trace length map. Minimum $P_{21}$ and $P_{22}$ were calculated using the maximum area and minimum fracture trace length map.

A source of uncertainty in mapping fracture intensities with distance from the crater is that one value represents a 2D area, covering a range of distances from the crater centre. The range of distances that a section covers depends on its orientation relative to the impact. Thin section planes that are roughly concentric to the crater centre have a smaller range of distances (~8 mm) than those oriented radially (up to 50 mm).

## 3. Results

Surface damage from the impact consists of a shallow, bowl-shaped impact crater which is truncated by material loss along a stepped surface from one corner of the block. The loss of material is substantial: $3.812 \times 10^5$ mm$^3$, ~12% of the block's initial volume (outlined in Figure 2). Surface fracture traces with macroscopic apertures are present on the remaining stone, with radially oriented traces centred on the crater, and traces sub-parallel to the target face (XY plane) up to 80 mm from the crater centre (Figure 2). Most radial fractures intersect the edge of the block and are visible on adjacent sides. The damaged surface, excluding the crater, has a stepped morphology with distinct steeply and gently dipping surfaces (Figures 1c and 2). The facet data shows two distinct orientations, one dipping steeply towards [X$_{min}$, Y$_{max}$] and the other sub-parallel to the XY plane. The degree of clustering of poles to fractures ranges from 10σ to 18σ, where σ is the number of standard deviations from sampling a random distribution.

The impact resulted in a shallow, bowl-shaped crater directly below the impact (Figure 2). The crater has a deep central pit surrounded by a shallow dipping region separated by a change of slope, illustrated on the top edge of the thin section drawing in Figure 1d, where the black arrow indicates spall fractures below an incipient fragment. The floor of the crater has a lighter colouration than surrounding damage as a result of the com-

minuted material and grain fracturing. BSE images from within 5 mm of the crater centre display fractures going through and around grains, conchoidal quartz fracture surfaces, and comminuted material (Figure 3a). The fractures observed can be seen penetrating the stone surface, where they have apertures < 20 μm. Around 20 mm from the crater centre, circumgranular, and to a lesser extent intragranular, fracturing is visible within the shallower spall zone, but the majority is circumgranular fracturing that separates grains from the clay matrix, leading to distinct oval-shaped depressions where grains have been 'plucked' from the surface (Figure 3b). Some fractures visible in BSE images cut across clay minerals at a high angle to mineral cleavage, similarly, observed in thin sections from below the surface (Figures 3b(ii) and 4i).

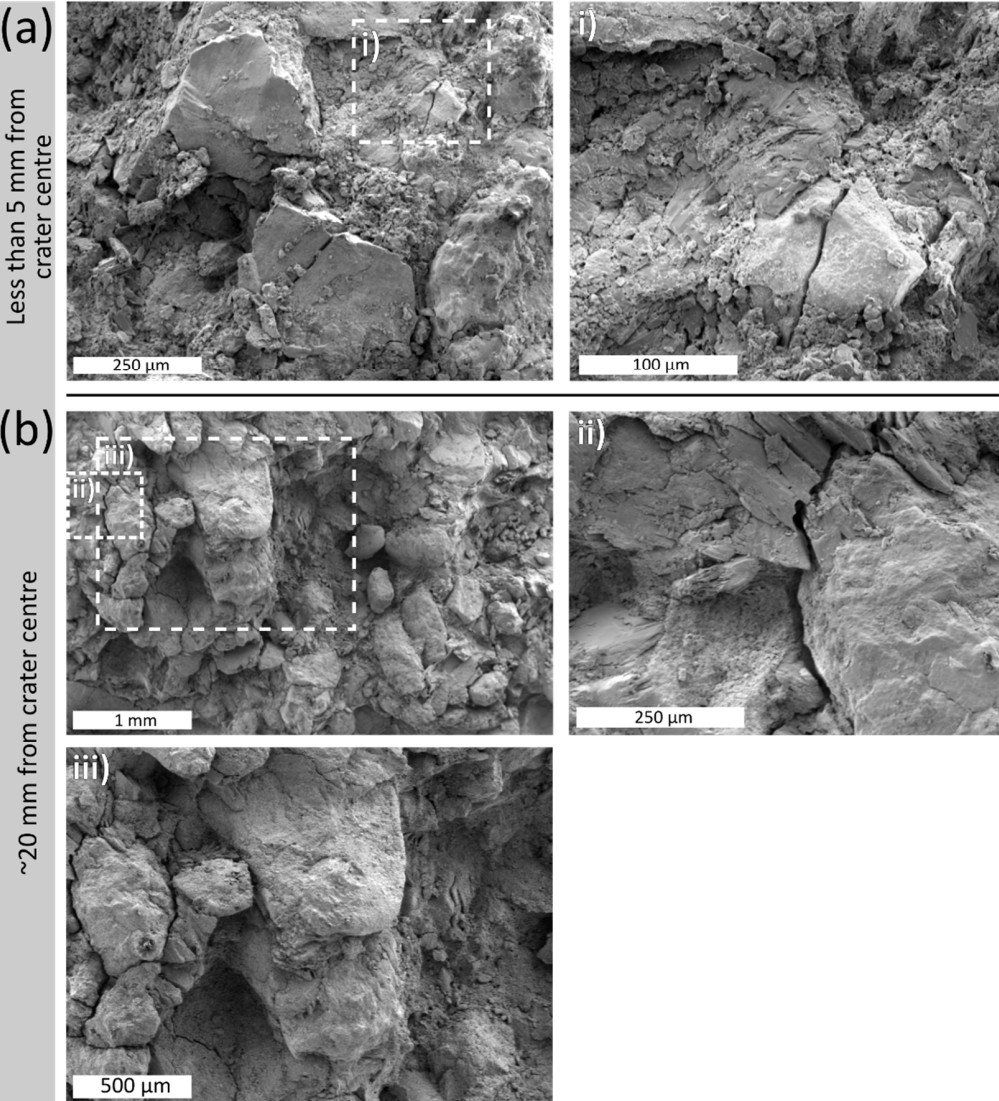

**Figure 3.** Backscatter electron (BSE) image of surface damage within the impact crater. (**a**) HS_IC is sampling the crater centre, showing heavily comminuted material, conchoidal fracture surfaces on quartz grains, and intragranular fracture paths (**i**). (**b**) Sample HS_CR from the spall zone of the impact crater shows grain plucking, less comminution, fracturing of clay minerals at a high angle to cleavage (**ii**), and a larger proportion of fractures having circumgranular paths around grains (**iii**).

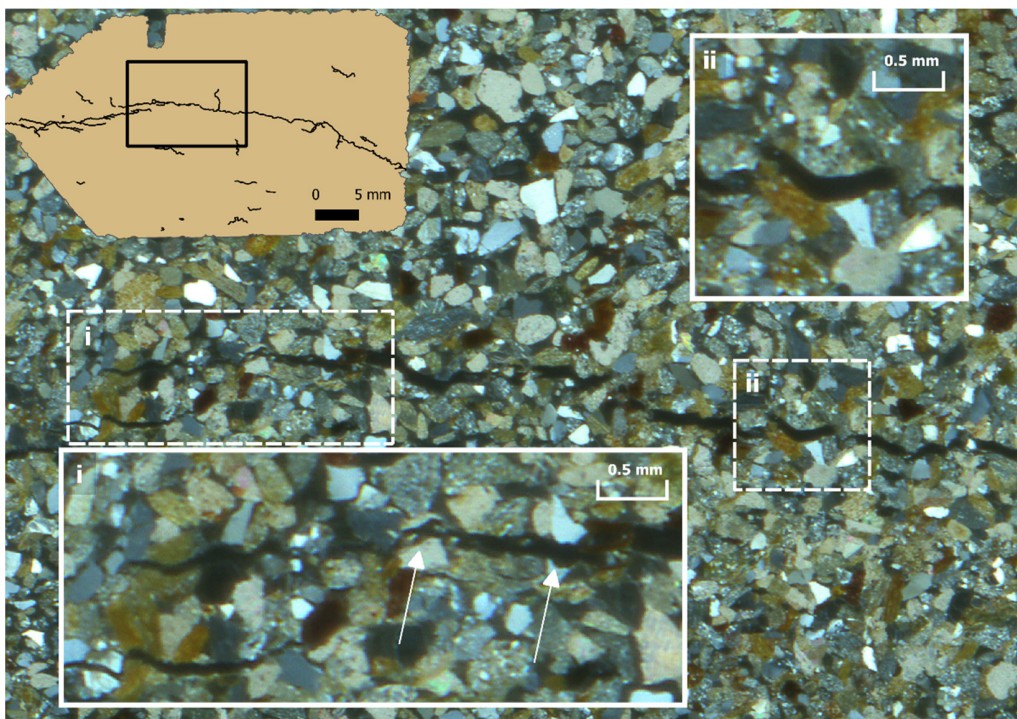

**Figure 4.** Thin section micrograph under cross polarised light of Sample HS_RF_1P showing the path of radially oriented fracture. The fracture path is both circum- and intragranular as seen in inset (**i**) and (**ii**) respectively. White arrows indicate intragranular fracturing. Note the fractures cutting across clay minerals at a high angle to cleavage in the lower left of (**i**), as well as the zone of overlap between the shorter fracture strands that make up the macro-scale radial fracture.

Radial fractures appear as a single trace at the macro scale (e.g., HS_RF_1P), but at the microscale are multiple shorter branches that overlap or join together (Figure 4). The aperture varies along the fracture length, narrowing at the fracture tips and overlap zones, and widening in the middle. Fracture paths are both circum- and intragranular. Sections close to the impact crater have open, curved fractures sub-parallel to the crater floor, linked by occasional short fractures with an approximately radial orientation (Figure 1d). Fracture paths are again indiscriminate between within grains and along grain-matrix boundaries. With increasing distance from the crater centre, fractures tend towards circumgranular paths and intragranular fractures are less common, particularly those traversing quartz grains.

Quantification of the fracture networks suggests that fracture intensity ($P_{21}$ and $P_{22}$) decreases linearly with increasing distance from the crater centre (Figure 5). The $P_{21}$ value of sample HS_RF_1P appears to differ from this trend and has a lower value (0.117 vs. 0.193) than sample HS_FS_4P which is 25 ($\pm$2) mm further from the crater. With the exception of HS_RF_2P, the characteristic branch length of samples ($L_c$) is approximately 1 mm (Figure 5). The impact has generated a combined 312,980 mm$^2$ of new internal and external surface area. The minimum estimate of the internal area is half that of external fracture surfaces (Table 1). The impact crater has a relatively small contribution to the overall induced damage, with the majority of the generated surface area related to internal and external fracturing, with a minimum estimate of 0.110 for $P_{32}$ intensity.

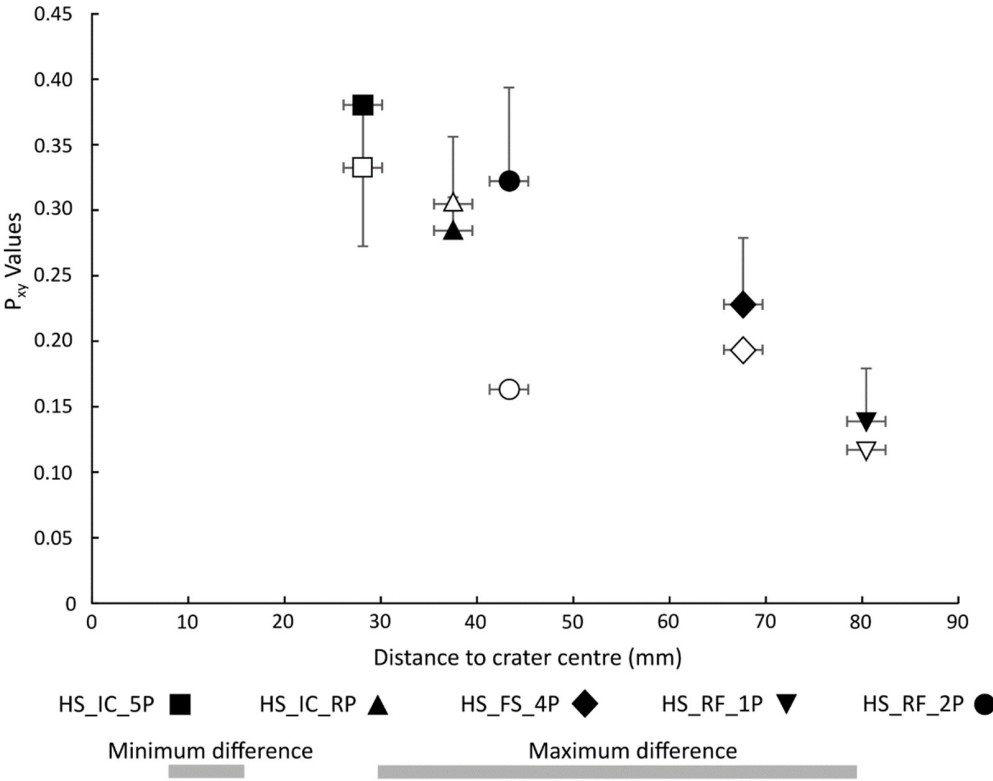

**Figure 5.** Graph showing the decrease in $P_{21}$ (hollow symbols) and $P_{22}$ (filled symbols) values with distance from the point of impact. The minimum and maximum differences show how much distance a section can represent in a single value (see Table A1 in Appendix A for values).

**Table 1.** Surface area measurements obtained from the 3D model of the damaged Huesca sandstone.

| Region | Area (mm$^2$) |
|---|---|
| External of damaged block includes: | 122,510 |
| • Impact Crater | 2520 |
| • Stepped Region | 17,850 |
| Internal fractures (min. estimate) | 10,470 |
| Total surface area (min. estimate) | 132,980 |

## 4. Discussion

The surface damage represented by the impact crater and stepped region is linked to a network of subsurface fractures, which consists of circum- and intragranular fracture paths of varying apertures that decay in intensity with increasing distance from the crater centre. Surface observations within the impact crater and spall zone show a shift of fracturing towards circumgranular pathways with increased distance from the impact, which is also seen throughout the subsurface sections and reflected in the fracture intensity plots. The micro-fractures provide evidence to support Gilbert et al.'s [7] suggestion that increased permeability and decreased surface hardness associated with the impact crater is related to micro-fracturing, as well as mirroring observations of grain fracture proximal to impact by Mol et al. [42]. Gilbert et al. [7] observed a light powdery appearance on the crater floor and a smaller surface hardness reduction relative to other areas of the impact damage. This is indicative of grain crushing and compaction directly below the impact, supported by this study's SEM observations of fractured grains and comminuted material, observations also made in hypervelocity impacts. Zones of pervasive fracturing and crushing are evidenced as impact breccia beneath natural craters [43] and as heavily comminuted grains in experimental samples [19,20]. Further similarities to hypervelocity experiments are

the bowl-shaped crater, the shallow surrounding spall zone, and the penetrative radial fractures [18,19,44]. Greater fracture intensity values closer to the crater centre, and direct observation of surface and subsurface fractures support observations of a decreasing degree of grain size reduction with distance from the impact by Buhl et al. [44]. The irregular fracture paths present across grains and along grain boundaries are similar to dynamic fractures where propagation stabilises at high velocities, resulting in rough and irregular fracture surfaces [44,45].

During the dynamic fracture caused by impacts, higher strain rates tend to result in higher fracture intensity, as more flaws are required to fail in order to accommodate the high strain rate [46]. Buhl et al. [47] measured axial strain and modelled the axial strain rate below hypervelocity impacts in sandstone where they observed a very high strain rate directly beneath the impact, which rapidly decayed within 4–5 projectile diameters (~8 mm in their study). For this study, 4–5 projectile diameter would equate to a distance of 30–38 mm (using the widest diameter of the projectile). Because stubs were removed from the crater centre, fracture analysis could not be performed closer than 28 mm from the crater centre, so these results may only represent a small portion of the sample that experienced the highest strain. As such, inferences of damage directly below the impact are drawn with care, but direct observation of grain comminution and micro-fracturing on the surface suggest that fracture intensities may be higher in this region when taken together with the clear relationship between fracture intensity and distance from the crater centre. The decrease in fracture intensity values with distance is similar to the decay in strain rate observed beyond 8 mm by Buhl et al. [47].

Fractures are an important control of the mechanical properties of masonry and the long term susceptibility of heritage to weathering. They provide new pathways for moisture ingress, and their influence on stone properties (surface area, porosity and pore size distribution, compressive strength, and modulus of compressibility) facilitates further deterioration through salt crystallisation and frost cycles, potentially resulting in the loss of large fragments of material [48,49]. This link between fracture damage and deterioration was explored further by Lebedeva and Brantley [12], who found weathering fronts advanced faster in stone with smaller fracture spacing (greater intensities). This would suggest that regions proximal to the impact may experience the fastest advance of weathering deterioration, and should therefore have a higher priority in terms of conservation strategies.

Structure from Motion (SfM) is a relatively quick and easy field method for capturing morphology without imposing additional deterioration or damage. SfM requires minimal investment, needing only a digital camera and computer, whereas other methods of 3D model generation such as terrestrial laser or structure from light scanning may require specialist equipment and proprietary software. SfM has been useful in cataloguing heritage as a whole, and SfM from drone-based cameras has proven archaeological applications, including the study of inaccessible sites, such as high walls [50]. The quality of SfM models produced in this study was sufficient to characterise impact damage morphology and quantify fracture areas. The estimation of internal fracture area relies on fracture traces being present across changing relief on the model (e.g., on different sides of the block). Limited relief, e.g., when fragments are held in place by adjacent blocks, or where visual observation of block sides is obscured, will result in underestimates of fracture surface area. However, models still provide valuable information for conservation work with regard to fracture orientations and length. Radial fractures are observed reaching the edge of the block, and can travel along mortar bonds and destabilise larger sections of masonry beyond the impacted block [51].

Even within the same calibre, ammunition can be highly variable, depending on its purpose (e.g., armour piercing), which affects characteristics such as shape (e.g., spitzer, round nose, hollow point), and composition. These variables influence the damage caused by impact. Hypervelocity experiments [52] suggest that variables resulting in greater kinetic energy lead to larger crater volumes. The angle of impact also influences the amount of

kinetic energy transferred to the target, and where peak pressures are experienced. Pierazzo and Melosh [53] showed that shallower angles of impact generate peak pressures further down trajectory and closer to the surface than perpendicular impacts. The likelihood of perpendicular impacts in a conflict scenario is small, and conclusions drawn in this study may need modification to deal with oblique impacts, for which the impacts described here are a baseline in ongoing investigations.

Microscale observations through SEM and thin section samples have demonstrated a link between damage visible on the surface and damage within the subsurface. Thin sections enable the relationship between subsurface fracturing and the impact to be quantified, supporting previous suggestions and observations that damage is greater closer to the impact point [5,7,47]. The negative trend of $P_{xy}$ values with distance from the crater centre suggest negligible fracture intensities 115–120 mm from the crater centre, approximately 80% of the block's dimensions. Further experiments are needed to investigate if this value is a constant.

## 5. Conclusions

This study has shown that an experimental impact into natural stone can result in substantial material loss from cratering and from the expansion of a macro-scale fracture network intersecting the edge of the target block. The stepped surface of the fracture network has two distinct orientations: one sub-parallel to the target face and the other steeply inclined towards one corner. The crater is surrounded by penetrative radial fractures that reach adjacent sides, and fractures parallel to the target face up to 80 mm from the crater floor. The total crater area is substantially less than that of the stepped region, and indeed of the internal fractures. Surface cratering, which is commonly the most apparent feature of conflict damage, may not be the most important expression of damage, with fractures accounting for ~4–7 times as much damage by area.

On the micro scale, open aperture and grain boundary fractures are visible in thin sections on both the surface and within the target block. Directly below the crater floor sub-parallel open aperture fractures traverse grains and grain boundaries, transitioning to fractures primarily along grain boundaries with increasing distance from the crater floor. Fracture intensities measured from the sections show a decrease from $P_{21} = 0.33$ close to the impact to $P_{21} = 0.12$ further away, with values that become negligible towards the margins of the block. Subsequent weathering poses a greater risk to regions proximal to the impact than those further away. Integrating scales of observation and non-destructive testing has shown surface and subsurface fracture damage to be linked throughout the block, meaning surface damage provides a foundation for understanding the internal damage caused by bullet impacts.

**Supplementary Materials:** The following are available online at https://www.mdpi.com/article/10.3390/geosciences11090395/s1, Figure S1: Fracture map and cross polar photomicrograph of section HS_IC_5P, Figure S2: Fracture map and cross polar photomicrograph of section HS_IC_RP, Figure S3: Fracture map and cross polar photomicrograph of section HS_RF_1P, Figure S4: Fracture map and cross polar photomicrograph of section HS_RF_2P, Figure S5: Fracture map and cross polar photomicrograph of section HS_FS_4P, Figure S6: 3D render of damaged Huesca block and minimum estimate for internal fracture area (orange), Table S1: Fracture trace and facet orientation data.

**Author Contributions:** Conceptualisation, methodology, data visualization, formal analysis, investigation, data curation, writing—original draft, writing—review and editing, O.C.; conceptualization, methodology, writing—original draft, writing—review and editing, supervision, T.B.; conceptualisation, investigation, data curation, writing—original draft, writing—review and editing, supervision, funding acquisition, L.M.; conceptualisation, investigation, writing—original draft, writing—review and editing, O.G. All authors have read and agreed to the published version of the manuscript.

**Funding:** This research was funded by the Leverhulme Trust, grant number RPG-2017-408. The APC was funded by Cardiff University's Institutional Open Access Fund.

**Data Availability Statement:** All data used is available in the supplementary materials.

**Acknowledgments:** We would like to thank Ben Williams and two additional colleagues who wish to remain anonymous for their constructive comments and discussion, greatly improving early versions of this manuscript. We also thank the two anonymous reviewers for their valuable suggestions that further improved the manuscript.

**Conflicts of Interest:** The authors declare no conflict of interest.

## Appendix A

**Table A1.** Table summarising the $P_{xy}$ values and errors for all sections.

| Sample | $d_{min}$ (mm) | $d$ (mm) | $d_{max}$ (mm) | $L_c$ (mm) | | $P_{21}$ (mm$^{-1}$) | | $P_{22}$ | | $P_{32}$ (mm$^{-1}$) |
|---|---|---|---|---|---|---|---|---|---|---|
| HS_IC_5P | 16 | 28 | 49 | 1.146 | +0.016 −0.314 | 0.332 | +0.001 −0.005 | 0.380 | +0.001 −0.108 | - |
| HS_IC_RP | 29 | 38 | 49 | 0.933 | +0.006 −0.246 | 0.305 | +0.005 −0.005 | 0.284 | +0.003 −0.072 | - |
| HS_FS_4P | 57 | 68 | 79 | 1.178 | +0.023 −0.274 | 0.193 | +0.003 −0.002 | 0.232 | +0.004 −0.051 | - |
| HS_RF_1P | 50 | 80 | 100 | 1.185 | - −0.353 | 0.117 | +0.001 −0.001 | 0.134 | - −0.040 | - |
| HS_RF_2P | 46 | 43 | 54 | 1.975 | +0.010 −0.408 | 0.163 | - −0.003 | 0.322 | - −0.071 | - |
| Full block | - | - | - | - | - | - | - | - | - | 0.101 |

$d_{min}$ = distance from the closest point of the section to impact centre, $d_{max}$ = distance from the furthest point of the section to impact centre, $d$ = distance to section centre.

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
