# Peer review of "Surface and Subsurface Damage Caused by Bullet Impacts into Sandstone"

_geosciences, doi:10.3390/geosciences11090395_

Round 1

Reviewer 1 Report

I commend the authors on such a well-written and well-researched paper.

The attention to both micro and macro details related to bullet impacts was very welcomed. Too often studies like this focuses on one or the other, and I always see that as a detriment. But here, the authors engage in the broader picture as informed by the micro, and that is refreshing to see.

The methods seem sound and more than adequate. The authors used their expertise well, and obtained outside assistance when necessary. This is the mark of an open-minded researcher, and I appreciate the authors' efforts.

The graphics are intense. Good, but intense. They showcase the data well, and are not superfluous. Often times when reviewing I see images that have no apparent relation to the topic. Not so here. And the expanded explanatory captions are also welcomed. That the information in the figure captions is not repeated in the text is great. Less redundancy. Very good.

The references seem appropriate, though there are studies that center on non-destructive techniques associated with stone deterioration, and adding such references at the end of the sentence of line 52 might be a nice touch to let readers know such techniques exist. These references could include:

  • Pope, G. A., Meierding, T. C., & Paradise, T. R. (2002). Geomorphology's role in the study of weathering of cultural stone. Geomorphology47(2-4), 211-225.
  • Hatır, M. E., Ä°nce, Ä°., & Korkanç, M. (2021). Intelligent detection of deterioration in cultural stone heritage. Journal of Building Engineering44, 102690.
  • Allen, C. D., Ester, S., Groom, K. M., Schubert, R., Hagele, C., Olof, D., & James, M. (2018). A Geologic Assessment of Historic Saint Elizabeth of Hungary Church Using the Cultural Stone Stability Index, Denver, Colorado. In Urban Geomorphology (pp. 277-302). Elsevier.
  • Jo, Y. H., & Lee, C. H. (2011). Making method of deterioration map and evaluation techniques of surface and three-dimensional deterioration rate for stone cultural heritage. Journal of Conservation Science27(3), 251-260.

Aside from that, I see no other problems with this manuscript. It is well-written, well-researched, and well-explained. The science seems sound, the topic is timely (and useful and interesting!), and it represents a novel approach to assessing cultural stone deterioration.

I don't say this often, but this was a genuine pleasure to review. Especially as someone who studies cultural stone deterioration. It was as if I was reading an already-refereed and peer-reviewed article. Very much kudos to the authors for such a fine paper.

Author Response

Point 1: The references seem appropriate, though there are studies that center on non-destructive techniques associated with stone deterioration, and adding such references at the end of the sentence of line 52 might be a nice touch to let readers know such techniques exist.

Response 1: The recommended sources were added to the fourth paragraph of the introduction [13-16].

Reviewer 2 Report

Dear authors,

I have now read your article titled “Surface and subsurface damage caused by bullet impacts into sandstone”. I have a personal interest in both firearms and fracture mechanics so I was delighted when this came across my desk! Overall, I found this to be a very interesting and worthwhile study. The study is well justified, well thought out and the manuscript is easy to follow. I recommend some minor revisions prior to acceptance in Geosciences:

  • The similarity between the sandstone used in the study and the sandstones in conflict regions could have been compared in more detail in my opinion in terms of their composition, grain size etc. A detailed description of the block used in experiments is given but we are not provided with much information on the petrology of comparable blocks in conflict zones. I suggest providing a little more detail about the petrology of the sandstones in conflict zones.
  • Even within a particular calibre of projectile there exists many manufacturers and types e.g. hollow vs round nose etc. Can the authors speculate how such variation would impact the results? Is there any potential for follow up work?
  • Figure 2: Is it possible to add a scale bar to the photo of the fractured block.
  • In the paper the projectile appears to hit the sandstone block perpendicular to its surface. However, in real scenarios more oblique collisions are often going to occur. How would this impact damage zone? I suggest a little discussion on this in the manuscript

I look forward to seeing this paper published!

Author Response

Point 1: The similarity between the sandstone used in the study and the sandstones in conflict regions could have been compared in more detail in my opinion in terms of their composition, grain size etc. A detailed description of the block used in experiments is given but we are not provided with much information on the petrology of comparable blocks in conflict zones. I suggest providing a little more detail about the petrology of the sandstones in conflict

Response 1: A brief description of the Umm Ishrin sandstone (Petra, Jordan) that has sustained historic damage has been added to the first paragraph of the methods. We recognise this is not an active conflict zone, but it provides a safe analogue for potential fieldwork, while experiencing the same climatic conditions and deterioration.

Point 2: Even within a particular calibre of projectile there exists many manufacturers and types e.g. hollow vs round nose etc. Can the authors speculate how such variation would impact the results? Is there any potential for follow up work?

Response 2: A new paragraph was added into the discussion section (para 5) to discuss briefly the effects of ammunition variability and angle of impact on the damage sustained. Additionally we now specify that the use of 90 degree impact in this study provides the baseline of comparison for ongoing investigations.

Point 3: Figure 2: Is it possible to add a scale bar to the photo of the fractured block.

Response 3: A scale bar has been added to Figure 2.

Point 4: In the paper the projectile appears to hit the sandstone block perpendicular to its surface. However, in real scenarios more oblique collisions are often going to occur. How would this impact damage zone? I suggest a little discussion on this in the manuscript.

Response 4: See response 2.